# UBR5 in Tumor Biology: Exploring Mechanisms of Immune Regulation and Possible Therapeutic Implications in MPNST

**DOI:** 10.3390/cancers17020161

**Published:** 2025-01-07

**Authors:** Diana Akinyi Odhiambo, Selina Fan, Angela C. Hirbe

**Affiliations:** Division of Oncology, Siteman Cancer Center, Washington University in St. Louis, St. Louis, MO 63110, USA; d.a.odhiambo@wustl.edu (D.A.O.); f.selina@wustl.edu (S.F.)

**Keywords:** neurofibromatosis type 1 (NF1), MPNST, ubiquitin proteosome system (UPS), UBR5, ubiquitin, HECT E3 ligases, immune checkpoint blockade (ICB), immunogenicity, immune evasion, immune “cold”, immune “hot”

## Abstract

Malignant peripheral nerve sheath tumor (MPNST) is an aggressive soft-tissue cancer with limited treatment options and poor survival rates, especially in cases with unresectable or metastatic disease. This review focuses on the role of *UBR5*, a gene frequently amplified in cancers and linked to immune suppression and tumor growth. Due to the typically low immune activity in MPNST, immune checkpoint blockade therapies—which have shown promise in treating metastatic cancers—are less effective. Studies in other cancers show that *UBR5* can influence immune responses and tumor progression, indicating it might play a similar role in MPNST. By exploring *UBR5*’s regulatory impact on the immune landscape, this review aims to provide insights into potential novel therapeutic strategies for inoperable and metastatic cases of MPNST.

## 1. Introduction

Malignant peripheral nerve sheath tumor (MPNST) is a rare but aggressive form of cancer, making up 5–10% of all soft-tissue sarcomas [1,2]. The prevalence of MPNST is one in 100,000 in the general population and affects both genders equally [1]. Approximately 50% of MPNST patients have Neurofibromatosis type 1 (NF1), while 10% have a history of radiation exposure, and the remaining cases are sporadic [1,3,4]. The average age of onset is between 30 and 50 years, but MPNST occurs around a decade earlier in patients with NF1 [1]. One of the most prevalent cancer predisposition syndromes, NF1 affects about 1 in 2500 individuals worldwide, and 8–13% of those with NF1 develop MPNST in young adulthood, often arising from benign tumors called plexiform neurofibromas (PN) [1,4,5].

Despite advancements in treatment, managing MPNST remains challenging. Surgical resection with wide negative margins remains the only curative treatment [2,6]. Unfortunately, factors such as tumor size, location, and metastasis complicate this approach [6,7]. The 5-year overall survival rate for NF1-associated MPNST ranges from 20% to 50% [5,7]. For patients with recurrent, unresectable, or metastatic disease, treatment options are limited, and outcomes are poor [6,7,8,9,10].

Clinical trials targeting signaling pathways involved in MPNST pathogenesis have not yielded significant clinical benefits, highlighting gaps in the understanding of the genetic and molecular underpinnings of this disease [2,6,8,11].

NF1-MPNST cases are characterized by *NF1* gene inactivation and loss of the *NF1*-encoded tumor suppressor protein, neurofibromin [2,6,8]. A negative regulator of Ras signaling, neurofibromin acts as a Ras-GTPase activating protein [2]. Therefore, the loss of neurofibromin leads to uncontrolled Ras activation and unregulated downstream signaling of pathways such as RAS/MEK/ERK and PI3K/AKT [2,12]. Nonetheless, given that only 10–13% of NF1 patients develop MPNST, biallelic loss of *NF1* alone is not sufficient to drive MPNST malignant transformation. Evidence shows that additional genetic alterations in the tumor suppressors *CDKN2A*, *TP53,* and *EED/SUZ12* are necessary for the transformation from benign PN to MPNST [6,13,14,15].

Given the limited efficacy of current regimens, there is growing interest in developing targeted therapies for MPNST. The focus has been on inhibiting Ras signaling or downstream pathways with the use of MEK and mTOR inhibitors. Unfortunately, none of these trials have reported clinical benefit for MPNST patients [2,6,8,9]. As such, there is a dire need for new therapeutic strategies and a deeper understanding of the molecular drivers of MPNST and the impact of the microenvironment on MPNST progression [6,16,17]. There is a growing appreciation for the role of immune, vascular, and stromal cells that form the MPNST microenvironment [17,18].

Immune checkpoint blockade (ICB) has transformed treatment in immune-inflamed cancers, such as melanoma, by providing a durable anti-tumor immune response [19,20,21]. The use of ICB in MPNST is overwhelmingly understudied [20]. The success of ICB is dependent on an immune cell-rich tumor microenvironment (TME) [19,20], a characteristic MPNST typically lacks, making them “cold” non-inflamed tumors [17,22]. As such, MPNST is unlikely to elicit a robust immune response following ICB [20]. Despite isolated reports of ICB response in individual MPNST patients [23,24,25], this lack of immunogenicity poses a significant challenge to the use of ICB in MPNST management [20]. Ubiquitin Protein Ligase E3 Component N-Recognin 5 (UBR5) is an E3 ubiquitin ligase commonly amplified and overexpressed in many cancers [26,27]. In cancer, UBR5 is involved in various cellular processes, including the cell cycle, DNA damage response, and the transcriptional machinery [28,29,30,31,32,33]. Overexpression of UBR5 is common in many cancer types, where it often promotes tumor development through mechanisms including but not limited to tumor suppressor degradation [26,34,35]. Similarly, UBR5 has been shown to influence the TME and immune response in Triple Negative Breast Cancer (TNBC) and ovarian cancer (OC). In a murine model of TNBC, genetic knockdown of *UBR5* was reported to facilitate antigen processing and presentation, triggering specific immune responses against tumors [30,36]. UBR5 also increased INFγ-mediated PDL1 transactivation, leading to immune evasion [30]. In a murine model of OC, UBR5 enhanced cancer growth and metastasis through the recruitment of immunosuppressive tumor-associated macrophages (TAMs) [37].

We have previously reported that chromosome 8 (Chr8) gain is a frequent and critical copy number alteration in MPNST and is associated with poorer overall survival and MPNST pathogenesis [38]. *UBR5*, one of the genes within the 8q arm [38], is often characterized by genetic alterations, particularly gene amplification in soft tissue sarcoma, including MPNST [38,39] [TCGA]. Evidence from studies into the role of *UBR5* in cancers such as TNBC and OC points to *UBR5* playing a key regulatory role in cancer biology and immune response [30,36,37]. While there is currently no published evidence linking *UBR5* to MPNST pathogenesis, by examining evidence of tumorigenic and immunoregulatory roles of *UBR5* in cancer, this review will explore mechanisms by which *UBR5* mediates immune regulation in cancer and the potential therapeutic implications of these mechanisms in MPNST.

## 2. The Ubiquitin Proteasome System and UBR5 Structure

Ubiquitin proteasomal system (UPS), a fundamental post-translational modification in eukaryotes, attaches ubiquitin (Ub) to protein substrates, thereby regulating various cellular processes such as protein degradation, signal transduction, apoptosis, immune response, and DNA repair [27]. While ubiquitination is commonly linked with protein degradation via the proteasome, ubiquitination also regulates protein levels, interactions, and localization [40,41]. UPS orchestrates this regulation through a cascade of three enzymes: Ub-activating enzyme E1, Ub-conjugating E2, and E3 ligases [40,41]. E3 enzymes, such as UBR5, interact directly with substrates, hence imparting substrate specificity to ubiquitination [42,43].

UBR5, a single polypeptide chain homologous to E6AP C terminus, belongs to the HECT-type E3 ubiquitin ligase family and is vital for mammalian embryonic development [27]. UBR5 is one of several E3 ligases encoded in the human genome. E3 ligases are classified into Ring between ring (RBR), the really interesting new gene (RING), and the homologous to E6AP C-terminus (HECT) family [27,29]. The modular structure of UBR5 comprises multiple domains (Figure 1), including a ubiquitin-associated (UBA) domain, a zinc-fingerlike ubiquitin-recognin (UBR) box, and an MLLE/PABP-interacting motif 2 (PAM2) binding domain adjacent to the HECT domain [29,44]. Structural studies outline the HECT domain as crucial for catalyzing ubiquitin transfer onto substrates. Notably, UBR5 is unique among N-end rule pathway E3 ligases for containing both an N-degron recognition UBR box and a HECT domain [29]. The N-degron recognition UBR box recognizes and degrades proteins based on the nature of their N-terminal amino acid residue [34]. Unlike other UBR-box containing E3 ligases, UBR5 is the only one containing a HECT domain, which regulates proteins by forming a thioester linkage between Ub and the conserved catalytic cysteine residue within the HECT domain [34]. Consequently, this transfers Ub from E3 ligase to target proteins [29,34,44].

Due to their giant size and often flexible architecture, only a dozen or so full-length E3 ligase structures have been reported. Until very recently, the full-length structure of UBR5 was not known. In 2023, Wang et al. determined the cryo-EM structure of the human full-length UBR5 at up to 2.66-Å resolution and revealed that UBR5 assembles into dimers and tetramers in solution [29]. Analysis revealed two crescent-shaped UBR5 monomers assemble head-to-tail to form the dimer, and two dimers bound face-to-face to form a cage-like tetramer with all four catalytic HECT domains facing the central cavity. In the dimer, the N-terminal region of one subunit and the HECT of the other form an “intermolecular jaw”. Although further work is needed to understand how oligomerization regulates UBR5 ligase activity, an important role for the jaw-lining residues in the recruitment of ubiquitin-loaded E2 to UBR5 was observed. Hodakova et al. also reported the structural characterization of UBR5, revealing a large helical scaffold decorated with numerous protein-interacting modules, giving UBR5 a range of possible mechanisms for substrate engagement [44]. Characterizing the functional implications of these modules, collectively suggested multivalent and possibly multimodal mechanisms of substrate recognition [29,44].

Structural characterization of UBR5 reveals key features that might allow for specificity in therapeutically targeting UBR5. The cryo-EM structure suggests UBR5 may have distinct functional roles depending on its oligomeric state (dimer vs tetramer). The “intermolecular jaw” involved in E2-Ub recruitment highlights potential sites for selective inhibition of Ub-mediated UBR5 activity. Additionally, the large helical scaffold with numerous protein-interacting modules described by Hodakova et al. suggests that UBR5 can engage substrates in a diversity of ways, further offering specificity. Reported multivalent binding offers multiple interaction points that can be selectively targeted to inhibit oncogenic substrates of interest. If validated, the proposed multimodal binding would allow UBR5 to recognize and bind substrates through various mechanisms. These could include recognizing different post-translational modifications, such as ubiquitination, or identifying distinct structural features. These structural insights offer a framework for creating small molecule inhibitors that are specific to domains or functional regions of UBR5, which could inhibit its oncogenic activity in cancer cells with precision while minimizing normal tissue toxicity.

Ubiquitin chain topology, including length and linkage type, dictates protein fate. For instance, branched ubiquitin chains, such as K11/K48, serve as potent signals for proteasomal degradation [40,41,45]. UBR5 plays a key role in assembling K11-K48 linked branched Ub chains, regulating the degradation of various substrates [24,39,40]. Structural analysis reveals that UBR5 specifically assembles K11-K48-linked branched Ub chains, including branching chains preformed at K11- or K63-linked chains [46]. UBR5 might collaborate with other E3 ligases to synthesize branched K48 linkages off preexisting ubiquitin chains. One such instance is UBR5, along with UBR4, which assembles K11/K48 branched ubiquitin chains, facilitating the rapid degradation of aggregation-prone proteins like 73Q-huntingtin (HTT) and C9orf7 [40,47].

Beyond its structural and functional role in ubiquitination, UBR5 has also been implicated in regulating normal cellular processes and contributing to both development and disease pathogenesis. A homolog of hyperplastic discs (HYD), *UBR5* plays critical tumor suppressor roles in *Drosophila melanogaster*, regulating cell proliferation during development [48]. Cells harboring mutated HYD failed to control proliferation, leading to tumorigenesis. Mutations in *HYD/UBR5* have been shown to lead to developmental abnormalities, including adult sterility secondary to germ cell defects [48].

In diseases like mantle cell lymphoma and Huntington’s disease, *UBR5* plays a crucial role in regulating normal cellular functions. For example, in mantle cell lymphoma, mutations that disrupt the HECT domain of *UBR5* impair normal B cell maturation and differentiation, potentially contributing to the transformation of cells into malignancies [45,49]. Similarly, in Huntington’s disease, although the mutant huntingtin protein (HTT) is typically overexpressed, studies using induced pluripotent stem cells (iPSCs) have shown that elevated levels of UBR5 can promote the degradation of the mutant HTT protein. This suggests that UBR5 might help to reduce the toxic effects of mutant HTT in disease models, offering potential insights into therapeutic strategies [45,47,48].

## 3. Functions of UBR5 in Cancer Biology

### 3.1. Role of UBR5 in DNA Damage Response (DDR)

#### 3.1.1. UBR5 and ATMIN Interaction

Zhang et al. demonstrated that upon irradiation, UBR5 ubiquitinates ATMIN, an ATM interactor, thus promoting ATM phosphorylation at DNA damage sites, allowing for efficient DNA repair initiation. UBR5-mediated ATM activation at damage sites favors cell survival after irradiation, driving radio-resistance. UBR5’s role in regulating the DNA damage response highlights its potential as a therapeutic target for enhancing response to DNA damage-inducing agents [27,50,51].

#### 3.1.2. UBR5 and TopBP1 Interaction

Previous studies have identified an interaction between UBR5 and DNA repair protein TopBP1. In vitro ubiquitination assays demonstrated that UBR5 ubiquitinates TopBP1 in the presence of the E2 enzyme UBCH4. TopBP1, a target for ubiquitination by UBR5, colocalizes with BRCA1 at stalled DNA replication forks, suggesting a role for UBR5 in regulating DNA replication and repair processes [52]. Additionally, UBR5-mediated ubiquitination of TopBP1 contributes to the activation of the DNA damage checkpoint kinase CHK2, further implicating UBR5 in DDR pathways [27,53]. Marcia munoz et al. demonstrated that UBR5 was necessary for CHK2-mediated G1/S and intra-S phase DNA damage checkpoint activation and for the maintenance of G2/M arrest after double-strand DNA breaks. Defective checkpoint activation in UBR5-depleted cells led to error-prone DNA synthesis, premature mitotic entry, accumulation of polyploid cells, and eventual cell death via mitotic catastrophe [54].

#### 3.1.3. UBR5 and TRIP12 Interaction

UBR5 collaborates with TRIP12 to suppress RNF168-mediated chromatin ubiquitination, hence preventing the spread of DNA damage signals away from the initial site of damage. TRIP12 and UBR5 limit the nuclear pool of RNF168, which functions to ubiquitinate a limited fraction of chromatin near double-stranded DNA breaks. The absence of TRIP12 and UBR5 results in the hyperaccumulation of RNF168, leading to aberrant chromatin ubiquitination and impaired DNA repair processes [27,55]. This interaction highlights the coordinated regulation of chromatin dynamics by UBR5 during the DNA damage response.

#### 3.1.4. UBR5 and Replication Fork Components Interaction

UBR5 additionally interacts with components of the replication fork, including the trans-lesion synthesis (TLS) polymerase, polη, to prevent erroneous recruitment of polη via ubiquitylated H2A. Depletion of UBR5 leads to erroneous polη recruitment and accumulation and consequent replication problems, such as slower S-phase progression and the accumulation of single-stranded DNA [56]. polη-mediated DNA replication inefficiency secondary to UBR5 loss emphasizes the significant role UBR5 plays in maintaining replication fork integrity and ensuring accurate DNA replication.

In summary, UBR5 plays diverse roles in both normal cellular functions and tumorigenesis. The unique interaction between UBR5 and key DNA damage-associated substrates, such as ATMIN, TopBP1, and replication fork components, highlights the significance of UBR5 in regulating DNA repair processes and maintaining genomic stability (Appendix A). A thorough review of the role of UBR5 in genomic stability is discussed by Shearer et al. [27]. There’s evidence that genomic instability and mutagenesis augment the number and diversity of tumor neoantigens, antigen presentation, and consequent, anti-tumor immune responses [57,58,59]. It is, therefore, fair to hypothesize that UBR5 deficiency in tumor cells may result in DNA repair deficiencies and consequent genomic instability. As such, further investigation into the intricacies of UBR5 functions and interactions may provide insights into novel therapeutic avenues for targeting the DDR and immune system.

### 3.2. UBR5 in Metastasis and Therapeutic Resistance

UBR5 has emerged as a critical player in cancer progression, metastasis, anti-apoptosis, and therapeutic resistance. The interactions of UBR5 with a diversity of substrates underpin its multifaceted activities that contribute to cancer cell survival and resistance to chemo and radiotherapeutics.

#### 3.2.1. UBR5 in Therapeutic Resistance and Anti-Apoptosis

Overexpression of UBR5 in tumors seems advantageous in the evasion of the cytotoxic effects of DNA-damaging agents. UBR5 drives this evasion by augmenting DNA repair machinery, cellular checkpoints, and the anti-apoptotic response. This is evidenced by UBR5-mediated degradation of the proapoptotic modulator of apoptosis protein 1 (MOAP-1) and modulator of apoptosis protein 1 (MOIP-1) in a HECT domain-dependent manner. UBR5-mediated degradation of MOAP-1 and MOIP-1 confers cisplatin resistance in ovarian cancer, illustrating the capacity for UBR5 to enhance cancer cell survival following chemotherapy [60]. Additionally, Matsuura et al., 2017 reported a UBR5-mediated transcriptional activation of the anti-apoptotic protein Mcl-1 in a manner that was independent of its E3 ubiquitin ligase activity. In this context, UBR5 influences apoptosis-associated proteins, Mcl-1, MOAP-1, and MOIP-1, through both ubiquitin ligase-dependent and -independent mechanisms. That UBR5 regulates Bcl-2 family proteins and their regulators underscores UBR5’s complex role in regulating apoptotic pathways and cancer cell survival. An additional instance in which UBR5 regulates the apoptotic machinery is evidenced by UBR5-mediated degradation of the proapoptotic regulator TXNIP through K48/K63-branched ubiquitin chains [61].

In an ERa+ breast cancer model, UBR5 overexpression induces tamoxifen resistance by upregulating β-catenin expression and activity. WNT/β-catenin signaling is known to drive the acquisition of stemness in various cancers. In a ubiquitin ligase-dependent manner, UBR5 led to β-catenin stabilization and consequent β-catenin driven therapeutic resistance. *UBR5* knockdown, conversely, decreased β-catenin nuclear accumulation without affecting β-catenin mRNA expression, suggesting a UBR5-mediated post-translational regulation of β-catenin [62]. The inhibition of UBR5/β-catenin signaling re-sensitized tamoxifen-resistant breast cancer in vivo. Similarly, UBR5-mediated β-catenin stabilization has also been described in a TNBC model of breast cancer [37].

In addition to modulating cancer stemness, UBR5 also modulates the levels of other critical proteins involved in cancer progression. UBR5 suppresses MYC expression levels through ubiquitination. MYC is a master regulator of malignant growth, but excess MYC protein levels prime both normal and cancer cells for apoptosis. In tumor cells, UBR5 fine-tunes MYC protein levels, balancing cell growth and survival, thus contributing to therapeutic resistance by protecting breast tumors from MYC-mediated apoptosis-priming [63]. Concordantly, UBR5 depletion in cells treated with replication inhibitors resulted in MYC-mediated apoptosis.

#### 3.2.2. UBR5 in Metastasis

The influence of UBR5 extends to metastasis and cancer cell growth. For instance, UBR5-mediated ubiquitination of the tumor suppressor CDC73 impacts lung colonization of metastatic cells. *UBR5* deficiency led to CDC73 upregulation, resulting in decreased metastatic tumor growth and increased sensitivity to apoptosis [64]. This relationship implicates UBR5 in the metastatic process and highlights its role in regulating apoptosis and cell proliferation pathways [50,51,65]. Additionally, Ziqi Yu et al. observed upregulation of *Trp53* mRNA following *Ubr5* deletion. Interestingly, this was reversed by silencing *Cdc73,* indicating a role for UBR5 in regulating the p53 pathway in part by degrading the tumor suppressor CDC73 [64].

In TNBC, UBR5 abrogation impairs angiogenesis within tumors, increases apoptosis, and induces growth arrest. The absence of *UBR5* disrupted mesenchymal-to-epithelial transition (MET) by deregulating E-cadherin expression, thus reducing the ability of micrometastases to establish within secondary organs [36]. Mei Song et al. elucidated further the mechanism by which UBR5 loss alters MET: *UBR5*^−/−^ cells displayed strongly reduced levels of E-cadherin, ID1, and ID3 mRNA and proteins all of which are critical regulators of mesenchymal–epithelial transition (MET) in metastatic colonization. UBR5 directly or indirectly regulated the transcription of *Id1* and *Id3* genes in a manner independent of its E3 ubiquitin ligase activity [37].

Furthermore, UBR5 promotes pancreatic cancer cell migration and invasion by degrading CAPZA1, a member of the F-actin capping protein family, in an E3 ubiquitin ligase-dependent manner. This degradation leads to F-actin accumulation and cytoskeletal remodeling, essential for cancer cell motility and invasion. The process highlights the role of UBR5 in regulating actin dynamics and promoting metastatic behavior [66].

In cervical cancer, UBR5 destabilizes TIP60, a lysine acetyltransferase involved in transcription, DNA damage response, and apoptosis. UBR5-mediated destabilization of TIP60 impairs its ability to acetylate p53, thereby suppressing p53-dependent transcriptional activation of genes involved in apoptosis and cell cycle arrest. This UBR5-mediated destabilization is driven by the HPV E6 oncogene to facilitate the oncogenic activity of the HPV E6 oncoprotein, which targets p53 for degradation, collectively promoting tumorigenesis and metastatic potential in cervical cancer cells [67].

Finally, in gastric cancer, UBR5 binds to and ubiquitinates gastrokine 1 (GKN1), a tumor suppressor that regulates cell growth and protects gastric mucosal integrity. The ubiquitination of GKN1 by UBR5 reduces its stability, leading to diminished GKN1-mediated suppression of cell proliferation and migration, thereby promoting cancer cell growth and metastasis [68]. Similarly, in colorectal cancer, UBR5 increases ubiquitination and degradation of ECRG4, a tumor suppressor, thus facilitating cell proliferation and survival [69]. These interactions illustrate the critical role UBR5 plays in deregulating tumor suppressors and enhancing oncogenic pathways. In glioma, the tumor suppressor ECRG4 is downregulated. ECRG4 inhibits the activity of NF-kB, suppressing the invasion, proliferation, and migration of glioma cells [66]. Qiang Wu et al. reported UBR5 upregulation in glioma tissues and cells, which promoted the migration and invasion of glioma cells by regulating the ECRG4/NF-kB pathway. Consequently, *UBR5* knockdown promoted the expression of ECRG4 and reduced the phospho-activation of NF-kB and phosphorylation of IkBa, a negative regulator of NF-kB [70].

In summary, UBR5 interacts with key proteins such as MOAP-1, MOIP-1, β-catenin, TXNIP, MYC, CDC73, CAPZA1, TIP60, GKN1, and ECRG4, NF-kB, highlighting its pivotal roles in promoting metastasis, therapeutic resistance, and cancer cell survival. Understanding these interactions provides critical insights into the potential of UBR5 as a therapeutic target in cancer treatment.

## 4. Role of UBR5 and Therapeutic Implications in MPNST

### 4.1. Role of UBR5 in Immune Modulation

Although chr8 gain has been reported in MPNST, the precise role played by the genes housed within this chromosome, including *UBR5*, is not fully understood. UBR5 has been implicated in various other cancers, underscoring its key role in tumorigenesis. Emerging evidence indicates that the tumorigenic activities of UBR5 are, in part, dependent on its modulation of the immune system. This is not surprising as components of the UPS, such as E3 ligases and deubiquitinases, have been discussed as modulators of both cellular and non-cellular components in the tumor microenvironment [71,72,73].

In a mouse model of OC, the loss of Ubr5 was found to enhance chemotherapy response and augment the immune response to anti-PD-1 and CAR T-cell therapy [37]. This model demonstrated that Ubr5 facilitated OC growth and metastasis by maintaining an immunosuppressive tumor microenvironment (TME). Ubr5 drove growth and immunosuppression through the recruitment and activation of Tumor-Associated Macrophages (TAMs) via paracrine signaling involving CCL2 and CSF-1. UBR5-mediated recruitment of TAMs contributed to an immunosuppressive TME that supports tumor growth and metastasis. Additionally, Ubr5 was essential for sustaining β-catenin signaling, which promoted cancer stemness and distant metastasis independently of TAMs, pointing to a tumor-autonomous process. The role of Ubr5 in modulating immune response is highlighted by the observed success of immune checkpoint blockade (ICB) and CAR T-cell therapy in this OC mouse model following the loss of *Ubr5*. This success is particularly noteworthy given the historically limited efficacy of immunotherapy in OC [74,75], underscoring the critical interplay between Ubr5 and the immune system in regulating tumor growth and Immunotherapy response.

In a murine model of Triple-Negative Breast Cancer (TNBC), Ubr5 also enhanced tumor growth in a manner dependent on the immune system. In *Ubr5*-deficient tumors, there was a notable increase in antigen presentation, the percentage of CD8+ T-cells and interferon-gamma (IFN-γ) producing CD4+ T-cells in the spleen, as well as an increased infiltration of polymorphonuclear leukocytes in UBR5 knockout tumors. Conversely, Ubr5 wild-type (WT) tumors exhibited a higher number of regulatory T-cells (Tregs) in the spleen and tumor-draining lymph nodes (TDLN), and a greater infiltration of mononuclear leukocytes was observed [36]. The proportion of tumor-infiltrating CD4 + and CD8 + T-cells was increased in *Ubr5^−/−^* tumor-bearing mice compared to control tumors. Furthermore, infiltrating CD8 + T-cells exhibited enhanced granzyme B expression, indicating a more active cytolytic state of the CD8+ T-cells in *Ubr5*-deleted tumors [37]. The infiltration of polymorphonuclear leukocytes in *Ubr5^−/−^* tumors was associated with reduced angiogenesis, suggesting a potential mechanism by which Ubr5 promotes tumor growth through immune modulation. Additionally, *Ubr5* depletion in tumor-bearing mice reduced surface levels of PD-L1 on TAMs, suggesting a role for Ubr5 in immune evasion [37].

The molecular mechanisms by which UBR5 regulates immune activation warrant further investigation. UBR5 has been implicated in β-catenin signaling, which has been shown to modulate immune cell function in a context-dependent manner. For instance, in melanoma, β-catenin signaling suppresses T-cell infiltration, contributing to immune checkpoint blockade (ICB) resistance [76,77,78]. UBR5-mediated stabilization of β-catenin may exacerbate these immune-exclusion mechanisms. Similarly, in non-small cell lung cancer (NSCLC), WNT/β-catenin signaling has been associated with immune-desert phenotypes, even in tumors with high tumor mutational burden (TMB), highlighting its role in shaping the immune microenvironment [79]. While a direct link between UBR5 and immune modulation through WNT/β-catenin signaling in NSCLC and melanoma remains to be established, the involvement of UBR5 in β-catenin stabilization suggests the potential involvement of UBR5 in these processes. Additionally, the involvement of UBR5 in CCL2/CSF-1 signaling, as reported in the TNBC model [37], indicates a broader impact of UBR5 on the cytokine milieu within the TME, influencing the recruitment and activation of various immune cell populations.

In another murine model of TNBC, UBR5 was found to regulate IFN-γ-mediated PD-L1 expression to promote tumor growth [30]. Tumor cells commonly exploit the PD-L1/PD-1 interaction to evade immune surveillance, a phenomenon associated with poor prognosis across various cancers [80]. PD-L1 expression can be induced by several cytokines, including IFN-γ, tumor necrosis factor-α (TNF-α), interleukins (ILs), and epidermal growth factor (EGF), via multiple signaling pathways such as JAK/STAT1/IRF1, NF-κB, PI3K/AKT/mTOR, and JAK/STAT3 [81].

In this TNBC model [30], UBR5 regulated IFN-γ-induced PD-L1 transcription in an E3 ubiquitin ligase-independent manner. Through its polyadenylate binding (PABC) domain, UBR5 enhanced PD-L1 transactivation by upregulating the transcription of the *EIF2AK2* gene that encodes protein kinase RNA-activated (PKR). The consequence was increased transcription of effectors, such as STAT1 and IRF1, downstream of PKR [30]. Further demonstrated in this same TNBC model, dual hairpin targeting of *UBR5* and PD-L1 (*CD274*) resulted in dramatically reduced tumor growth and lung metastasis and significantly extended survival. Along the same lines, UBR5 has been shown to stabilize the WNT/β-catenin signaling, a key driver of cancer stemness [37,63,76].

In melanoma, β-catenin signaling has been shown to mediate resistance to ICB therapy [76,77]. Underscoring the significant role of WNT/β-catenin signaling in immune evasion. Takeuchi et al. reported that NSCLC that lacked immune cell infiltration into the TME despite high tumor mutational burden (TMB), a marker of high immunogenicity, preferentially up-regulated the WNT/β-catenin pathway. In high TMB NSCLC, which are highly immunogenic, TMB does not always correlate with PD-L1 expression. In this NSCLC model, a PD-L1 independent immunosuppressive TME that is driven by WNT/β-catenin signaling was observed [79]. These additional pieces of evidence underscore the role of UBR5 in regulating oncogenic pathways such as WNT/β-catenin to drive tumor immune evasion.

While the role of UBR5 in sarcomas, including MPNST, is understudied, the oncogenic pathways discussed above, through which UBR5 regulates immune escape, have been implicated in MPNST pathogenesis. For instance, signaling downstream of EGFR is known to induce PD-L1 expression and contribute to immune evasion. Similarly, the PI3K-AKT-mTOR and WNT/β-catenin pathways play roles in PD-L1 induction and immune evasion [81]. These signaling pathways are not only crucial drivers of tumorigenesis in MPNST but are also therapeutic targets in MPNST clinical trials, albeit with limited clinical success [6,8,82]. MAPK-ERK2, a downstream target of the Ras signaling implicated in NF1-MPNST pathogenesis, is reported to regulate UBR5 function in cellular signaling through phosphorylation [83,84]. As such, exploring the interplay between UBR5 and these key oncogenic pathways driving MPNST could uncover mechanisms to enhance therapeutic response, in part through augmenting anti-tumor immune response.

### 4.2. UBR5 in MPNST: Therapeutic Potential

In our lab, we have extensively studied eight NF1-MPNST parental tumors and their PDX counterparts. Our research highlighted pronounced aneuploidy, including chr8 gain, in NF1-MPNST compared to the generally diploid PNFs. Using bulk RNA-seq, we identified *UBR5* among chr8 genes with the highest expression in MPNST [38]. Sarcomas typically exhibit more copy-number aberrations than other cancers, with MPNST showing particularly high CNAs [39,85]. This is a critical discovery, as DNA aneuploidy is recognized as an independent risk factor for decreased metastasis-free survival in sarcomas, regardless of histologic grade or lymphovascular invasion [39,86,87].

In dissecting the MPNST immune profile, Holand et al. reported DNA copy number gains, including 8q gain, which also correlated with immune deficiency in MPNST tumors [39]. Chr8q gain has been reported in Ewing sarcoma and other pediatric soft-tissue sarcomas, suggesting this may be a critical event in multiple sarcoma types, including NF1-MPNST [38]. Moreover, our data linked Chr8 gain with poor OS in soft-tissue sarcoma datasets from TCGA. Chr8q contains numerous cancer-critical genes, including *MYC* (8q24), which is well-studied across various cancers, including MPNST, where upregulation of *MYC* and or its targets was correlated with tumor recurrence and immune deficiency [39,88]. Additionally, 8q houses other significant cancer-related genes, including UBR5, whose role in posttranslational regulation of Myc and Myc-mediated apoptosis-priming has been reported [32,63]. Underscoring the significance of Chr8q gain in MPNST, Maren Holand et al. classified MPNSTs into two transcriptomic subtypes defined primarily by immune signatures and proliferative processes. “Immune deficient” MPNSTs were more aggressive, characterized by proliferative signatures and high genomic complexity (CNAs), including 8q gain.

Based on our study, among the significantly upregulated genes within chr8, genetic knockdown of *UBR5* significantly impacted MPNST-cell survival. Preliminary unpublished findings from our lab revealed that knockdown of *UBR5* in MPNST-cells led to reduced proliferative capacity, increased apoptotic cell death, and decreased tumor growth in vivo. These results, coupled with the established role of UBR5 in the pathogenesis of cancers such as ovarian, lung, and breast cancer, highlight potential involvement in MPNST progression.

## 5. MPNST Immune Architecture: Mechanisms of Immune Evasion

Although immune-based treatment strategies, such as ICB, have revolutionized cancer care in malignancies like melanoma and non-small cell lung cancer (NSCLC) [89,90], progress in applying these strategies to MPNST has been notably limited [20,91]. NF1 patients exhibit germline loss of a single *NF1* allele, resulting in *NF1* haploinsufficiency (*NF1^+/−^*) and hyperactive Ras signaling across all cells in the body, including immune cells. Somatic loss of the second *NF1* allele in Schwann cell precursors triggers plexiform neurofibroma formation. The *NF1^+/−^* MPNST TME conditions play a pivotal role in PN development and malignant transformation through the secretion of growth factors, chemokines, and proinflammatory mediators [17,18,92].

Mast cells are one such immune cell type in the neurofibroma microenvironment that plays a critical role in NF1-associated tumorigenesis. *Nf1^−/−^* Schwann cells (SCs) hypersecrete stem cell factor (SCF), the ligand for c-kit, which is essential for mast cell development and survival [16,18]. *Nf1* haploinsufficient (*Nf1^+/−^*) mas cells show enhanced migration, proliferation, survival, and degranulation in response to SCF in a Ras/PI3K-dependent manner [17,18]. Among the effector proteins hypersecreted by *Nf1^+/−^* mast cells is TGF-beta (TGFβ), promoting increased proliferation and collagen production by *Nf1^+/−^* fibroblasts, a hallmark of PNF formation [18]. Additionally, mast cells hypersecrete various pro-angiogenic growth factors, including VEGF and metalloproteinases (MMPs), potentially contributing to PNF tumorigenesis [84,85,86]. This tumorigenic TME characteristic has motivated ongoing clinical trials investigating KIT/CSF1R inhibitors, targeting neurofibromas associated with mast cell infiltration [1,17,18]. Similarly, Hirbe et al. and Dodd et al. demonstrated that the *Nf1* haploinsufficient TME accelerated MPNST onset in genetically engineered mouse models (GEMMs) of MPNST [14,92]. Unsurprisingly, an enriched myeloid cell population, particularly elevated mast cells, was observed in these tumors. Consistent with these preclinical findings, a phase I clinical trial evaluating a combination strategy targeting mast cells and TAMs in MPNST yielded promising results. The combination of pexidartinib (KIT, CSF1R, and FLT3 inhibitor) and sirolimus, an mTOR inhibitor (NCT02584647), demonstrated efficacy in 12 out of 18 patients with advanced sarcoma, including five with MPNST. Median progression-free survival (PFS) and OS in the MPNST cohort were 18.6 weeks and 145.1 weeks, respectively [93].

Despite these advances, other immune strategies such as ICB, ICB combined with targeted therapy, CAR-T-cell therapy, and oncolytic viruses are still in their infancy in MPNST [91,94]. Nonetheless, the promotion of tumor formation and growth by mast cell-induced inflammation highlights the involvement of the MPNST TME in tumor pathogenesis and highlights the potential to restore anti-tumor immune function therapeutically. A summary of ongoing clinical evaluation of these immune strategies is provided in Table 1.

Emerging clinical data highlight the potential of PD-1/PD-L1 checkpoint blockade in MPNST, particularly in tumors expressing high levels of PD-L1. In clinical case reports, PD-L1 expression was observed in 70–100% of tumor cells in three patients and 2% in one patient, as determined by IHC staining [23,24,25,91,98]. Case reports of these four MPNST patients, who were initially refractory to conventional therapies but achieved complete responses with pembrolizumab or nivolumab, further underscore the potential of ICB in MPNST management.

Discrepancies between the success of these individual responses and broader ICB-related clinical outcomes MPNST may be attributed to various factors. Highlighted in these reports is the need for robust biomarkers to identify which MPNST patients are likely to derive clinical benefit from ICB. Additional factors such as inter- and intra-tumoral heterogeneity, initial treatment prior to ICB, and variable PD-L1 expression levels among patient cohorts may also contribute to this discrepancy [99]. In a study of MPNST transcriptomes pooled from Gene Expression Omnibus (GEO), 30% of MPNST samples had scores predictive of ICB response. This study used a validated predictive score model based on T-cell exclusion and dysfunction signatures to assess the extent of predicted response to ICB. Concomitant with existing evidence of heterogeneity, considerable variability in the behaviors of immune cells, even within TME of MPNST tumors with the same CTL score, was observed [100].

PD-L1 expression has been considered the best available predictive biomarker for ICB response. However, it has notable limitations, as not all patients with elevated PD-L1 levels exhibit positive responses to ICB therapy [101,102]. As such, efforts to identify predictive and prognostic biomarkers aim to enhance our understanding of the complex interplay between cancer and T-cell response [91]. In MPNST, potential biomarkers for immunotherapy have not yet been evaluated in large-scale standardized prospective trials. Nonetheless, preclinical evidence demonstrates the extent of immune evasion in MPNST. For instance, immune genes such as HLA, which encode MHC class I and II, the transcription factor MHC II transactivator (MHC2TA), the transporter associated with antigen processing (TAP1), and the related chaperone CD74, show reduced expression in NF1-associated tumors compared to normal human Schwann cells highlighting downregulation of or impaired antigen presentation processes in MPNST [103].

Biomarkers for future trials in MPNST will need to highlight evidence, or lack thereof, of immune activation and intact antigen presentation. Some relevant candidate biomarkers in MPNST include PD-L1copy number gain and expression [103], TILs [91,99], and TMB [91]. Typically, MPNSTs are characterized by low PD-L1 expression; however, reports on PD-L1 expression in MPNST are inconsistent, with some studies reporting high expression and others reporting low PD-L1 expression [20]. Elizabeth Shurell et al. reported that 13% of MPNST tissue microarray lesions had at least 5% PD-L1 staining. In these samples, MPNST was characterized by absent PD-1 expression, low PD-L1 expression, although higher than benign neurofibroma, and significant CD8+ TIL presence [104]. Similarly, transcriptomic analysis of T-cell signatures revealed that a majority of human MPNST (62 out of 73 samples profiled) exhibited an immunologically “cold” phenotype characterized by reduced cytotoxic T-cell infiltrates and increased tumor immune dysfunction and exclusion (TIDE) scores [100]. These findings highlight the lack of robust immune cell infiltration in MPNST, which is necessary to elicit a robust anti-tumor immune response, emphasizing the challenges to the effective application of immunotherapy approaches in this malignancy.

On such a basis, MPNST has indeed been described as a non-inflamed/immune cold [99,100,104]. Spatial gene expression profiling of human PNSTs across the neurofibroma-MPNST continuum revealed that atypical neurofibromatous neoplasm with uncertain biologic potential (ANNUBP) exhibited enhanced signatures of antigen presentation and T-cell infiltration, while MPNST was characterized by immune-excluded TME with an increased expression of genes associated with immune exhaustion [39,105]. Interestingly, “benign” neurofibromas contiguous to MPNST similarly harbored distinct gene expression profiles characterized by signatures of impaired antigen presentation. Of particular interest in this study is the potential to utilize signatures of impaired antigen presentation to identify precursor lesions histopathologically classified as “benign” at high risk of undergoing malignant transformation [39,105].

An intriguing question is what oncogenic mechanisms underlie this immunosuppressive environment that appears indispensable for the malignant transformation of ANNUBP to MPNST. Changes in immune composition resulting from the loss of the PRC2 complex—a histone-modifying complex primarily responsible for maintaining transcriptional silencing through H3K27 trimethylation (H3K27me3)—represent a plausible hypothesis. Indeed, evidence suggests that loss of the PRC2 complex, a frequent occurrence in MPNST, is linked to reduced antigen presentation and altered immune signatures [4,13,39]. PRC2 loss is also associated with an aggressive disease course, proliferative signatures, and frequent gains of chr8q and triploid genomes in MPNST [4,39]. In a recent study classifying MPNSTs into two transcriptomic subtypes defined primarily by immune signatures and proliferative processes, 8q was reported among the most frequently affected by copy number gain among the “immune deficient” tumors (69% and 66%, respectively). Importantly, possible target genes, including *UBR5,* were identified, among others, on chr8q.

Similarly, PRC2 inactivation resulted in ICB resistance through the reprogramming of the chromatin landscape, disrupting chemokine production and impairing antigen presentation and T-cell priming [10]. Consistent with these findings, Cortes-Ciriano et al. reported by whole genome sequencing, coupled with transcriptomic and methylation profiling of 95 NF1-related tumors, a significant association between H3K27 trimethylation (H3K27me3) status and immune-phenotype [4]. H3K27me3 loss was strongly correlated with decreased infiltration of immune cells into the TME, downregulation of granzyme expression, and decreased activation of adaptive immunity. H3K27me3 retention, on the other hand, was associated with an immune-cell-rich phenotype [4].

Tumors with a high tumor mutational burden (TMB) are likely to harbor more neoantigens. As such, albeit with some exceptions, higher TMB correlates positively with response to ICB in multiple cancers [21], with a definition of high TMB being >10 mutations per megabase [106]. Soft tissue sarcomas, in general, are reported to have a low TMB of 1–2.5 mutations/Mb [107]. However, in an analysis of 100,000 different cancers, 8.2% of MPNSTs had mutations of more than 20 mutations/Mb [108]. In a separate analysis of whole-exome sequencing data of 12 MPNST patient samples, somatic coding variants per tumor ranged from 7 to 472 with a median value of 63 [109].

While immunotherapy, particularly ICB therapy, has revolutionized the treatment of several malignancies, its efficacy in MPNST remains limited. The immunosuppressive tumor microenvironment driven by *NF1* haploinsufficiency and other pro-tumorigenic factors, such as PRC2 loss, WNT/β-catenin, and Ras/PI3K signaling, pose significant challenges to effective immune-based treatments. Promising clinical trials targeting mast cells and TAMs, [(pexidartinib: KIT, CSF1R, and FLT3 inhibitor + sirolimus, an mTOR inhibitor) (NCT02584647)], underscore the potential for combination strategies, yet the immunosuppressive TME and variable response to PD-1/PD-L1 blockade, highlight the need for predictive biomarkers and strategies to augment MPNST immunogenicity. A deeper understanding of the genetic and molecular mechanisms underlying MPNST, including the role of Chr8q gain and *UBR5* amplification in shaping the immune architecture and antitumor immune response, will be crucial for overcoming immune evasion and improving outcomes in MPNST.

## 6. Attempts to Enhance MPNST Immunogenicity: Prospects for ICB

Curative treatment for MPNST typically involves surgical resection with negative margins, but this approach is often constrained by factors such as tumor size and proximity to nerves [6]. For patients with recurrent, unresectable, or metastatic disease, treatment options are limited, and outcomes are poor [6,7]. ICB therapy has shown promise in the treatment of inoperable and undruggable malignancies, suggesting potential benefits for MPNST patients [6,7]. As discussed previously, the success of ICB is dependent on an immune-rich TME. Unfortunately, evidence highlights MPNST as a non-inflamed “immune cold” malignancy characterized by low immune cell infiltration, posing a significant challenge to the use of ICB therapy. Nonetheless, reports of anti-PD-1/PD-L1 success, albeit few, have prompted efforts to increase intratumoral T-cell density to enhance MPNST immunogenicity.

A recent study harnessed the cGAS/STING/IFN pathway to augment the innate immune response [20]. The cytosolic DNA–sensing enzyme cGAS binds to dsDNA and initiates a cascade of events leading to the production of type I IFNs and proinflammatory cytokines and chemokines. This cytokine milieu presumably recruits immune cells into the TME. Using a GEMM of MPNST (cisNf1^+/−^ p53^+/−^), the STING agonist reprogrammed the TME to enhance T-cell infiltration and sensitized tumors to ICB therapy. Treatment of MPNSTs with a STING agonist caused activation of the STING pathway, upregulation of cytokines and chemokines, and infiltration of immune cells, including T-cells, into the tumor. Consequently, enhanced tumor growth delay and significant apoptosis were observed upon STING activation followed by ICB therapy compared to STING agonists alone [20].

Other strategies have employed the use of oncolytic viruses. Recent reports have highlighted the potential of viral treatments to enhance immune infiltration in MPNSTs. Ghonime et al. demonstrated that a multimodal oncolytic virus engineered to express EphrinA2—an antigen found in various tumor types—can elicit a strong immune therapeutic response in immune-competent mouse models of both glioma and MPNST [110]. Additionally, a more recent study showed that intratumoral delivery of an inactivated modified vaccinia virus Ankara (MVA) could transform the immune desert environment in MPNSTs, promoting immune infiltration and hence rendering tumors more responsive to ICB therapy [111]. In addition to these reports, further preclinical studies evaluating oncolytic virus therapy in MPNST are reviewed here [91].

A 2023 study demonstrated that dual targeting of Ras effector kinases, CDK4/6 and MEK, sensitized MPNST to ICB therapy. Dual inhibition of CDK4/6 and MEK demonstrated synergistic antitumor activity in patient-derived xenografts (PDX) and de novo tumors in immunocompetent mouse models of MPNST. PD-L1 blockade alone delayed tumor growth but caused little tumor regression compared to combination CDK4/6-MEK and PD-L1 inhibition. Resistant tumors were characterized by an immunosuppressive TME with elevated MHC II-low macrophages and increased tumor cell PD-L1 expression. On the other hand, in responsive tumors, CDK4/6-MEK inhibition led to an increase in plasma cells and cytotoxic T-cell infiltration compared to resistant tumors [112]. These findings are particularly noteworthy given that tumor-infiltrating plasma cells and B cells (TIL-Bs), along with the antibodies they produce, are emerging as key components of the immune response against cancer [113]. Notably, plasma cell tumor infiltration has not been reported with the CDK4/6-MEK drug combination or the single agents in any tumor model, including MPNST [112]. Emerging evidence suggests that intratumoral plasma cells may serve as a better predictor of ICB response in cancer patients [113,114]. TIL-Bs are believed to use a distinct mode of antigen presentation that fosters the creation and maintenance of immunologically ‘hot’ tumor microenvironments involving T-cells, myeloid, and natural killer cells. Additionally, by attenuating self-tolerance mechanisms, TIL-Bs minimize the extent of immune editing, a process tumors adopt to evade immune surveillance [115,116,117].

These preclinical studies, among others (Table 2), highlight potential avenues to reprogram the MPNST immune desert into a more immunologically active TME. The role of UBR5 in immune modulation, as already discussed, presents another promising avenue for further research. Evidence from other cancers indicates that UBR5 promotes tumor growth by maintaining an immunosuppressive tumor microenvironment and facilitating immune evasion mediated by WNT/β-catenin and PD-L1, among other pathways. Targeting UBR5 could enhance the immunogenicity of MPNSTs, improving their response to immune checkpoint blockade. Exploring the interaction between UBR5 and the tumor microenvironment in MPNST will provide valuable insights for developing effective immunotherapeutic strategies for this therapeutically challenging malignancy.

## 7. Conclusions

The aggressive and metastatic nature of MPNST, coupled with limited effective treatment options, calls for novel therapeutic strategies to improve patient outcomes. To this end, preclinical and clinical studies have focused on the inhibition of the RAS/MEK/ERK and PI3K/AKT/mTOR pathways, which play a crucial role in MPNST tumorigenesis and progression, although these trials have resulted in minimal clinical benefit [2,6,8].

There is a growing appreciation for the role of MPNST TME in the formation and transformation of PNFs into MPNST. The role of RAS signaling and *NF1* haploinsufficient TME in PNF formation and malignant transformation has prompted the studies of immune and combinatorial strategies, including targeting mast cells and TAMs. Notably, the phase I clinical trial combining pexidartinib with the mTOR inhibitor sirolimus and its success thereof in MPNST trial patients supports this approach [16,92,93].

Isolated patient cases, albeit limited, have reported success with ICB therapy in patients with elevated tumor PD-L1 expression, suggesting potential for ICB therapy, particularly in MPNST patients with inoperable or metastatic disease [23,24,104]. However, the immune cold nature of MPNST presents a challenge to ICB therapy, emphasizing the need for strategies to enhance MPNST immunogenicity [20,22,99,104,111].

Genetic knockdown of the E3 ubiquitin ligase, *UBR5*, in preclinical models of cancer, has shown promise in attenuating tumor growth and enhancing anti-tumor immune response through mechanisms including increased antigen processing and presentation, elevated T-cell infiltration, reduced TAMs recruitment, and reduced PDL-1-mediated immune evasion [30,36,37]. Although further research is needed to substantiate the combined targeting of UBR5 and ICB therapy, the scattered success of ICB therapy, coupled with evidence of UBR5’s role in immune evasion, suggest that this combination could address the unmet therapeutic needs of MPNST patients with recurrent, unresectable, or metastatic disease. 

## Figures and Tables

**Figure 1 cancers-17-00161-f001:**
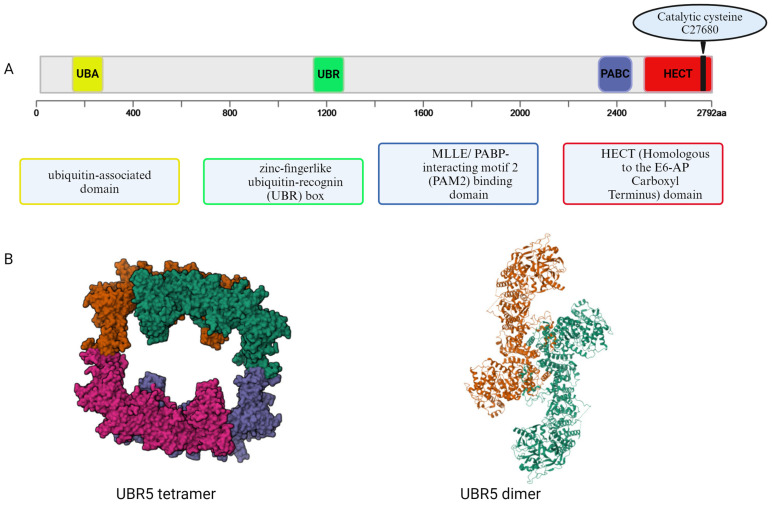
(**A**) Schematic Representation of the UBR5 Protein Domains: The UBR5 protein contains several functional domains: a ubiquitin-associated (UBA) domain (yellow), a zinc-finger-like ubiquitin-recognin (UBR) box (green), an MLLE/PABP-interacting motif 2 (PAM2) binding domain (blue), and a HECT (Homologous to the E6-AP Carboxyl Terminus) domain (red) containing the catalytic cysteine (C2768). Created with Biorander. (**B**) Structural Visualization of UBR5 Complexes: **Left Panel**: UBR5 tetramer (PDB ID: 8EWI) visualized as a surface model showing the quaternary assembly of four UBR5 monomers. **Right Panel**: UBR5 dimer (PDB ID: 8D4X) visualized as a ribbon model depicting the dimerization of two UBR5 molecules. Structures were retrieved from the RCSB Protein Data Bank (RCSB.org) with respective DOIs: https://doi.org/10.2210/pdb8EWI/pdb and https://doi.org/10.2210/pdb8d4x/pdb.

**Table 1 cancers-17-00161-t001:** Immune-based strategies in MPNST: clinical trials.

No.	Study Name	Phase	Agent	Clinical Trial No.	Modality	Status/Findings	Source
1.	PLX3397 Plus Sirolimus in Unresectable Sarcoma and Malignant Peripheral Nerve Sheath Tumors (PLX3397)	I & II	PLX3397, Sirolimus	NCT02584647	Small-Molecule Inhibitors	Active, not recruiting; patient enrollment: 43; interventional model: parallel assignment. Clinical benefit was observed in 12 out of 18 (67%) evaluable subjects, with three partial responses (all in TGCT) and nine stable disease. Tissue staining indicated a decreased proportion of activated M2 macrophages within tumor samples with treatment [93].	[91]
2.	Vaccine Therapy in Treating Patients With Malignant Peripheral Nerve Sheath Tumor That Is Recurrent or Cannot Be Removed by Surgery	I	Edmonston strain measles virus genetically engineered to express neurofibromatosis type 1 (oncolytic measles virus encoding thyroidal sodium-iodide symporter [MV-NIS])	NCT02700230	Oncolytic virus	Completed; patient enrollment: 9; interventional model: single-group assignment; no results available.	[91]
3.	Neoadjuvant Nivolumab Plus Ipilimumab for Newly Diagnosed Malignant Peripheral Nerve Sheath Tumor	I	nivolumab and ipilimumab	NCT04465643	Immune modulators (PD-1 and CTLA-4)	Recruiting; patient enrollment: 18; interventional model: single-group assignment.	[91]
4.	A Study of APG-115 in Combination With Pembrolizumab in Patients With Metastatic Melanomas or Advanced Solid Tumors	Ib/II	APG-115 and pembrolizumab	NCT03611868	Immune modulator (PD-1) and small molecule inhibitor	Recruiting interim results show stability in 53% of the MPNST cohort when treated for >4 cycles.	[91]
5.	B7H3 CAR-T-cell Immunotherapy for Recurrent/Refractory Solid Tumors in Children and Young Adults	I	Biological: second generation 4-1BBζ B7H3-EGFRt-DHFR (B7H3-specific CAR-T-cells and CD19 specific CAR-T-cells) biological: second generation 4-1BBζ B7H3-EGFRt-DHFR (selected) and a second generation 4-1BBζ CD19-Her2tG	NCT04483778	CAR-T-cells	Active, not recruiting.	[91]
6.	EGFR806 CAR-T-cell Immunotherapy for Recurrent/Refractory Solid Tumors in Children and Young Adults	I	Biological: second generation 4-1BBζ EGFR806-EGFRt biological: second generation 4-1BBζ EGFR806-EGFRt and a second generation 4 1BBζ CD19-Her2tG	NCT03618381	CAR-T-cells	Recruiting.	[91]
7.	Donor Stem Cell Transplant After Chemotherapy for the Treatment of Recurrent or Refractory High-Risk Solid Tumors in Pediatric and Adolescent-Young Adults	II	Allogeneic hematopoietic stem cell transplantation	NCT04530487	Stem cell transfer	Terminated; patient enrollment: 1; interventional model: single-group assignment; no results available.	[91]
8.	Nivolumab and Ipilimumab in Treating Patients With Rare Tumors	II	Nivolumab and Ipilimumab	NCT02834013	Immune modulators (PD-1 and CTLA-4)	Active, not recruiting.	[91]
9.	A Study of Pembrolizumab in Patients With Malignant Peripheral Nerve Sheath Tumor (MPNST), Not Eligible for Curative Surgery	II	Pembrolizumab	NCT02691026	Immune modulators	Terminated; patient enrollment: 8; interventional model: single-group assignment; no results available.	
10.	Lorvotuzumab Mertansine in Treating Younger Patients With Relapsed or Refractory Wilms Tumor, Rhabdomyosarcoma, Neuroblastoma, Pleuropulmonary Blastoma, Malignant Peripheral Nerve Sheath Tumor, or Synovial Sarcoma	II	Lorvotuzumab mertansine	NCT02452554	Antibody-drug conjugates	Clinical activity limited [95].	
11.	BLESSED: Expanded Access for DeltaRex-G for Advanced Pancreatic Cancer, Sarcoma and Carcinoma of Breast	I/II	DeltaRex-G	NCT04091295	Tumor targeted gene therapy, human cyclin G1 inhibitor	Available case report: A 14-year-old female with MPNST of the parotid gland and lung metastases. Previous treatments included chemotherapy (doxorubicin, ifosfamide, temozolomide, sorafenib) and immunotherapy with interleukin-2. She received intravenous Rexin-G for 2 years. Outcome: no evidence of active disease [96].	
12.	HSV1716 in Patients With Non-Central Nervous System (Non-CNS) Solid Tumors	I	HSV1716	NCT00931931	Oncolytic virus	Completed. Recruitment of patients from the first cohort (3) was completed without DLT or procedure-related severe adverse events. Two patients from the second cohort have been treated and have not shown any DLT or procedure-related severe adverse events [97].	
13.	MASCT-I Combined With Doxorubicin and Ifosfamide for First-line Treatment of Advanced Soft Tissue Sarcoma	II	MASCT-I (dendritic cells and effector T-cells injections) with Doxorubicin and Ifosfamide	NCT06277154	MASCT-I	Not yet recruiting.	
14.	Nivolumab and BO-112 Before Surgery for the Treatment of Resectable Soft Tissue Sarcoma	I	BO-112, and BO-112 with nivolumab for patients going through preoperative radiotherapy for definitive surgical resection	NCT04420975	dsRNA immunotherapy	Active, not recruiting.	

**Table 2 cancers-17-00161-t002:** Immune-based strategies in MPNST: preclinical studies.

No.	Study Name	Agent	Modality	Findings
1.	Treatment of orthotopic malignant peripheral nerve sheath tumors with oncolytic herpes simplex virus	Oncolytic herpes simplex viruses (oHSVs)	Oncolytic virus	Oncolytic HSV G47Δ showed efficacy and safety in immunodeficient and immunocompetent MPNST models. A single intratumoral injection of G47Δ inhibited tumor growth and prolonged survival significantly. HSV G47Δ expressing platelet factor 4 injection showed prolonged survival. HSV G47Δ expressing interleukin (IL)-12 showed significantly improved efficacy of the virus [118].
2.	Aurora A kinase inhibition enhances oncolytic herpes virotherapy through cytotoxic synergy and innate cellular immune modulation	Alisertib and HSV1716	Oncolytic virus	The alisertib and HSV1716 combination therapy showed higher antitumor efficacy compared to each monotherapy. In two xenograft models of MPNST and neuroblastoma, the administration of the combination decreased tumor growth and increased survival [119].
3.	Dominant-Negative Fibroblast Growth Factor Receptor Expression Enhances Antitumoral Potency of Oncolytic Herpes Simplex Virus in Neural Tumors	Oncolytic HSV with dominant-negative FGF receptor (dnFGFR)	Oncolytic virus	In vitro, bG47Δ-dnFGFR demonstrated greater tumor death in murine MPNST 61E4 compared to the control vector bG47Δ-empty. In vivo, bG47Δ-dnFGFR demonstrated greater efficiency at inhibiting tumor growth and angiogenesis in 3-18-4 MPNST tumors in nude mice compared to bG47Δ-empty [120].
4.	Oncolytic HSV Armed with Platelet Factor 4, an Antiangiogenic Agent, Shows Enhanced Efficacy	Oncolytic HSV with insertion of transgene platelet factor 4 (PF4)	Oncolytic virus	PF4 transgene insertion into oncolytic HSV G47Delta showed more effective inhibition of tumor growth and angiogenesis in vivo in mouse 37-3-18-4 malignant peripheral nerve sheath tumor model than non-expressing parent bG47Delta-empty [121].
5.	Oncolytic HSV and Erlotinib Inhibit Tumor Growth and Angiogenesis in a Novel Malignant Peripheral Nerve Sheath Tumor Xenograft Model	oHSV mutants (G207 and hrR3) with erlotinib (EGFR inhibitor)	Oncolytic virus	oHSV injection showed higher antitumor activity than erlotinib. Combination therapy of oHSV and erlotinib demonstrated a trend towards increased antiproliferation. EGFR inhibition did not reduce oHSV efficacy [122].
6.	Molecular analysis of human cancer cells infected by an oncolytic HSV-1 reveals multiple upregulated cellular genes and a role for SOCS1 in virus replication	G207 oHSV	Oncolytic virus	Upregulation SOCS1 correlated with human cancer cell increased sensitivity to G207 oHSV. Decreased levels of SOCS1 lowered virus replication [123].
7.	Molecular engineering and validation of an oncolytic herpes simplex virus type 1 transcriptionally targeted to midkine-positive tumors	oHSV fused with human MDK promoter to the HSV type 1 neurovirulence gene, γ134.5	Oncolytic virus	oHSV-MDK-34.5 administration decreased tumor growth and increased medical survival in vivo in MPNST tumor-bearing nude mice [124].
8.	Oncolytic measles virus as a novel therapy for malignant peripheral nerve sheath tumors	Engineered MV Edmonston vaccine strain with human sodium iodide symporter	Oncolytic measles virus therapy	Inhibition of tumor growth in MPNST tumor xenograft and increased long-term survival in mice were observed after a single MV-NIS intratumoral administration. For treated mice that survived to the end, all except one had no substantial-sized tumor [125].
9.	STING activation reprograms the microenvironment to sensitize NF1-related malignant peripheral nerve sheath tumors for immunotherapy	Activating simulator of IFN genes (STING) signaling with ICB	ICB + tumor microenvironment	Treatment with STING agonist sensitized tumors to ICB and showed increased apoptotic cell death in mouse genetic and human xenograft MPNST models [20].

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
