# Peer review of "UBR5 in Tumor Biology: Exploring Mechanisms of Immune Regulation and Possible Therapeutic Implications in MPNST"

_cancers, 2025, doi:10.3390/cancers17020161_

Round 1

Reviewer 1 Report

Comments and Suggestions for Authors

1. Please provide a figure presenting the URB5 pathway and its role in DNA Damage Response

2. Please provide a figure presenting the NF1 signaling pathway and URB5 pathway - how do these intersect?

3. Please prepare a figure presenting the MPNST microenvironment, immune cells in MPNST tumor, and the role of URB5 in the regulation of immune response

4. Please cite appropriate data on MPNST treatment in the introduction, including:

https://www.tandfonline.com/doi/full/10.2217/fnl-2018-0026

https://www.mdpi.com/2072-6694/16/19/3266

https://journals.viamedica.pl/oncology_in_clinical_practice/article/view/60061

https://www.esmo.org/guidelines/guidelines-by-topic/sarcoma-and-gist/clinical-practice-guidelines-soft-tissue-and-visceral-sarcomas

Author Response

We appreciate the opportunity to revise and resubmit our manuscript “UBR5 in the Tumor Microenvironment: Exploring Mechanisms of Immune Regulation and Possible Therapeutic Implications in MPNST” for potential publication in Cancers.

Based on the insightful comments of the two expert reviewers, we have modified our review. We have addressed each comment in a point-by-point manner below.

Reviewer 1

Comment 1: Please provide a figure presenting the URB5 pathway and its role in DNA Damage Response

Figure 1: Legend

Role of UBR5 in maintaining genomic integrity. Ub=Ubiquitin

Comment 2: Please provide a figure presenting the NF1 signaling pathway and URB5 pathway - how do these intersect?

Response:

We greatly appreciate the reviewer’s suggestion to illustrate the NF1 and UBR5 signaling pathways and their intersection. At present, there is no direct evidence of a physical or functional interaction between UBR5 and NF1 in the context of NF1-mutant cancers, such as MPNSTs. However, we hypothesize that these pathways may intersect through crosstalk involving downstream signaling components, particularly those regulated by receptor tyrosine kinases (RTKs).

UBR5 has been implicated in the regulation of RTK effector pathways, including RAS/RAF/MEK/ERK and PI3K/AKT/mTOR, which are critical to NF1-MPNST pathogenesis in the context of NF1 loss. Specifically:

  • UBR5 and WNT/β-catenin signaling: UBR5 stabilizes β-catenin, a downstream effector of RTKs and RAS signaling, which is often hyperactivated in cancers.
  • UBR5 and mTOR signaling: UBR5’s regulation of protein turnover has been linked to components of the mTOR pathway, which is frequently dysregulated in NF1-mutant cancers.
  • Genomic instability: UBR5 deficiency has been associated with genomic instability and altered cellular stress responses, which could hypothetically influence NF1-driven signaling or immune evasion mechanisms.

These potential intersections remain hypothetical, primarily derived from studies in other cancer types rather than direct experimental evidence in NF1-driven MPNSTs. While it is conceptually feasible to speculate on these interactions, we currently lack the experimental data to generate a figure accurately depicting this intersection. Nonetheless, we agree that this is an important area of investigation and a core question driving my ongoing PhD research.

Comment 3: Please prepare a figure presenting the MPNST microenvironment, immune cells in MPNST tumor, and the role of URB5 in the regulation of immune response

Response:

We thank the reviewer for their thoughtful suggestion to provide a figure illustrating the MPNST microenvironment, the immune cells involved, and the role of UBR5 in regulating immune responses. We have included a graphical abstract (shown above) that addresses these aspects in detail.

This figure highlights the key features of the immune microenvironment in NF1-mutant MPNSTs, contrasting "immune-cold" tumors with poor immune infiltration and response to immunotherapy, against "immune-hot" tumors with enhanced immune activity and therapeutic responsiveness. The following points demonstrate how the figure aligns with the reviewer’s request:

  1. Comprehensive Overview of the Microenvironment: The figure visually represents the major cellular players in the tumor microenvironment (TME), including CD8+ T cells, NK cells, regulatory T cells (Tregs), tumor-associated macrophages (TAMs), and cancer cells, along with their respective roles in immune suppression or activation.
  2. UBR5’s Role in Immune Regulation: The figure integrates UBR5 into the conceptual framework of immune regulation, specifically in its hypothesized role in transitioning from an immune-cold to an immune-hot TME. While specific mechanisms remain under investigation, UBR5 is depicted as a key player influencing immune activation, antigen presentation, and suppression of immunosuppressive cell types.
  3. Alignment with MPNST Pathophysiology: By illustrating immune evasion mechanisms (e.g., exclusion of CD8+ T cells, impaired antigen processing), the figure contextualizes how UBR5 deficiency or dysregulation might contribute to altered immune responses in MPNSTs.

We believe this graphical representation sufficiently addresses the reviewer’s comment by summarizing the immune dynamics of the MPNST microenvironment and highlighting the hypothesized involvement of UBR5 in regulating these processes.

Comment 4. Please cite appropriate data on MPNST treatment in the introduction, including:

https://www.tandfonline.com/doi/full/10.2217/fnl-2018-0026

https://www.mdpi.com/2072-6694/16/19/3266

https://journals.viamedica.pl/oncology_in_clinical_practice/article/view/60061

https://www.esmo.org/guidelines/guidelines-by-topic/sarcoma-and-gist/clinical-practice-guidelines-soft-tissue-and-visceral-sarcomas

Addressed: Lines 56/57, 59/60, 72/73

Reviewer 2 Report

Comments and Suggestions for Authors

The review by Odhiambo et al is a detailed survey on the various roles of UBR5, a HECT-type E3 ubiquitin ligase, in tumor biology and their therapeutic implications in different cancer types with a focus on malignant peripheral nerve sheath tumor (MPNST). Overall, this is a well-written and informative review. The promising therapeutic utility of UBR5 in cancer is well described. I suggest the following revisions for improvement.

-       The title and the abstract seemed to prepare the reader for a focused review on UBR5 on the immune landscape of the tumor microenvironment in MPNST. However, more than half of the review is a general overview on the roles of UBR5 in tumor biology. I suggest at least revising the title and the abstract to include other aspects of the review.

-       The overall numbering structure/organization of the review is not clear. I suggest dividing the review into 3 major parts: (1) the ubiquitin proteasomal system and UBR5 structure, (2) function of UBR5 in cancer biology (with subsections on regulation of DNA damage repair, metastasis/therapy resistance and immune modulation), (3) role of UBR5 and therapeutic implications in MPNST. The subsection 2 on the function roles of UBR5 (lines 146-160) in normal tissues in the current version can be combined with the subsection 1 on the structure and function of UBR5.

-       I suggest including a figure on the protein domain structure of UBR5 in the section on UBR5 structure.

-       In the section on UBR5 in metastasis, it is unclear how the interaction between UBR5 and TIP50 in cervical cancer (lines 273-275) and between UBR5 and GKN1 in gastric cancer (lines 275-280) promote metastasis.

-       The authors mentioned the potential role of WNT/b-catenin signaling in immune modulation in several cancer types (triple negative breast cancer, melanoma and non-small cell lung cancer). It is not clear from the cited findings how UBR5 interacts or regulates the WNT/b-catenin pathway to affect altered immune cell function/landscape in these cancer types? The authors cited the role of UBR5 in stabilizing WNT/b-catenin signaling (lines 353-354) but given the diverse roles of this pathways in cancer biology, the interaction might be cancer type/context dependent, without necessarily affecting the biology of tumor-infiltrating immune cells.

-       Table 1 includes status/findings on the immune-based clinical trials in MPNST. The current version includes findings on some completed trials. I suggest including brief findings on all the completed trials. To save space, the ‘source’ (or reference) column can include the references in numbers.

-       I suggest including major findings on the pre-clinical studies (e.g. effects on tumor growth, immune cell types/molecules affected) in table 2. Again, to save space, the ‘source’ column can include the references in numbers.

-       Some abbreviations in the review should be spelled out when first used (e.g. OC in line 301, TDLN in line 322, NSCLC in line 358 and ANNUBP in line 498)

-       A reference is missing in line 553.

-       There are a few typographical errors (chr8 amplification? In line 294, NF11haploinsufficiency in line 621).

Author Response

We appreciate the opportunity to revise and resubmit our manuscript “UBR5 in the Tumor Microenvironment: Exploring Mechanisms of Immune Regulation and Possible Therapeutic Implications in MPNST” for potential publication in Cancers.

Based on the insightful comments of the two expert reviewers, we have modified our review. We have addressed each comment in a point-by-point manner below.

Reviewer 2

 Comment 1: The title and the abstract seemed to prepare the reader for a focused review on UBR5 on the immune landscape of the tumor microenvironment in MPNST. However, more than half of the review is a general overview on the roles of UBR5 in tumor biology. I suggest at least revising the title and the abstract to include other aspects of the review.

Addressed:

  • Title
  • Abstract: Lines 27-29

Comment 2: The overall numbering structure/organization of the review is not clear. I suggest dividing the review into 3 major parts: (1) the ubiquitin proteasomal system and UBR5 structure, (2) function of UBR5 in cancer biology (with subsections on regulation of DNA damage repair, metastasis/therapy resistance and immune modulation), (3) role of UBR5 and therapeutic implications in MPNST. The subsection 2 on the function roles of UBR5 (lines 146-160) in normal tissues in the current version can be combined with the subsection 1 on the structure and function of UBR5.

Addressed:

The Ubiquitin Proteasome System and UBR5 structure: 125, 135-150, 152-166, 177-192

  •       I suggest including a figure on the protein domain structure of UBR5 in the section on UBR5 structure.

Figure 2: Legend:

  1. Schematic Representation of the UBR5 Protein Domains: The UBR5 protein contains several functional domains: a ubiquitin-associated (UBA) domain (yellow), a zinc-finger-like ubiquitin-recognin (UBR) box (green), an MLLE/PABP-interacting motif 2 (PAM2) binding domain (blue), and a HECT (Homologous to the E6-AP Carboxyl Terminus) domain (red) containing the catalytic cysteine (C2768).
  2. Structural Visualization of UBR5 Complexes:
  • Left Panel: UBR5 tetramer (PDB ID: 8EWI) visualized as a surface model showing the quaternary assembly of four UBR5 monomers.
  • Right Panel: UBR5 dimer (PDB ID: 8D4X) visualized as a ribbon model depicting the dimerization of two UBR5 molecules. Structures were retrieved from the RCSB Protein Data Bank (RCSB.org) with respective DOIs: https://doi.org/10.2210/pdb8EWI/pdb and 10.2: https://doi.org/10.2210/pdb8d4x/pdb

Panel A created with BioRender.

Comment 3: In the section on UBR5 in metastasis, it is unclear how the interaction between UBR5 and TIP50 in cervical cancer (lines 273-275) and between UBR5 and GKN1 in gastric cancer (lines 275-280) promote metastasis.

Addressed: Lines 306-317

Comment 4: The authors mentioned the potential role of WNT/b-catenin signaling in immune modulation in several cancer types (triple negative breast cancer, melanoma and non-small cell lung cancer). It is not clear from the cited findings how UBR5 interacts or regulates the WNT/b-catenin pathway to affect altered immune cell function/landscape in these cancer types? The authors cited the role of UBR5 in stabilizing WNT/b-catenin signaling (lines 353-354) but given the diverse roles of this pathways in cancer biology, the interaction might be cancer type/context dependent, without necessarily affecting the biology of tumor-infiltrating immune cells.

Addressed: Lines 371-382

Comment 4: Table 1 includes status/findings on the immune-based clinical trials in MPNST. The current version includes findings on some completed trials. I suggest including brief findings on all the completed trials. To save space, the ‘source’ (or reference) column can include the references in numbers.

-       I suggest including major findings on the pre-clinical studies (e.g. effects on tumor growth, immune cell types/molecules affected) in table 2. Again, to save space, the ‘source’ column can include the references in numbers.

Both tables addressed

Comment 5: Some abbreviations in the review should be spelled out when first used (e.g. OC in line 301, TDLN in line 322, NSCLC in line 358 and ANNUBP in line 498)

Addressed: Lines 92/93, 362, 377, 544/545

-       A reference is missing in line 553.

Addressed: Line 599

-       There are a few typographical errors (chr8 amplification? In line 294, NF11haploinsufficiency in line 621).

Addressed: Lines: 334, 668
